# Modulation of Kv Channel Gating by Light-Controlled Membrane Thickness

**DOI:** 10.3390/biom15050744

**Published:** 2025-05-21

**Authors:** Rohit Yadav, Juergen Pfeffermann, Nikolaus Goessweiner-Mohr, Toma Glasnov, Sergey A. Akimov, Peter Pohl

**Affiliations:** 1Institute of Biophysics, Johannes Kepler University Linz, 4040 Linz, Austria; rohit.yadav@jku.at (R.Y.); juergen.pfeffermann@jku.at (J.P.); nikolaus.goessweiner-mohr@jku.at (N.G.-M.); 2Institute of Chemistry, Karl-Franzens-University, 8010 Graz, Austria; toma.glasnov@uni-graz.at; 3Frumkin Institute of Physical Chemistry and Electrochemistry, Russian Academy of Sciences, Moscow 119071, Russia; akimov_sergey@mail.ru

**Keywords:** Kv channel, voltage sensor, photoswitchable lipids, electrophysiology

## Abstract

Voltage-gated potassium (Kv) channels are e ssential for shaping action potentials and rely on anionic lipids for proper gating, yet the mechanistic basis of lipid–channel interactions remains unclear. Cryo-electron microscopy studies suggest that, in the down state, arginine residues of the voltage sensor draw lipid phosphates upward, leading to a local membrane thinning of ~5 Å—an effect absent in the open state. To test whether membrane thickness directly affects voltage sensor function, we reconstituted Kv channels from *Aeropyrum pernix* (KvAP) into planar lipid bilayers containing photoswitchable lipids. Upon blue light illumination, the membrane thickened, and KvAP activity increased; UV light reversed both effects. Our findings indicate that membrane thickening weakens the interaction between lipid phosphates and voltage-sensing arginines in the down state, lowering the energy barrier for the transition to the up state and thereby promoting channel opening. This non-genetic, membrane-mediated approach provides a new strategy to control ion channel activity using light and establishes a direct, reversible link between membrane mechanics and voltage sensing, with potential applications in the remote control of neuronal excitability.

## 1. Introduction

Voltage-gated ion channels enable rapid cellular excitability, particularly within neuronal systems. Among them, voltage-gated potassium ion (Kv) channels play a crucial role in regulating the shape and duration of action potentials [1,2]. Kv channels are tetrameric proteins, each subunit containing a voltage-sensing domain (VSD) formed by four transmembrane segments (S1–S4) arranged around a central conducting pore domain composed of two transmembrane helices (S5 and S6) contributed by each subunit [3]. Kv channels respond to changes in membrane voltage by conformational changes, [4] enabling pore opening and closing through the movement of positively charged amino acid residues in S4, known as gating charges [5,6]. Due to these gating charges, VSDs exhibit pronounced interactions with the surrounding phospholipid headgroups, which can induce local deformations in the lipid membrane that may influence channel function [7,8,9]. Such lipid bilayer deformations have been extensively studied and characterized [8,10,11,12], revealing that lipids may modulate channel function and dynamics. The presence of non-phospholipids has been shown to impair channel activation by voltage, while variations in lipid shape significantly influence voltage-dependent behavior [13]. Additionally, enzymatic hydrolysis of the head group in sphingomyelin was reported to inhibit the activity of Kv2.1 channels, further highlighting the critical role of lipid head groups for Kv channel function [14]. Furthermore, changes in bilayer mechanics induced by alkanols also affect KvAP conductance and activation kinetics [15].

Interestingly, Kv channels are sensitive to membrane tension [16]. One proposed explanation is that membrane tension favors a state of the protein with an increased cross-sectional area within the membrane plane. Kv channels fit this description, as they are much more compact in their closed state than in their open state [17]. However, it seems doubtful that the area expansion modulus of the protein is compatible with this idea, given that the surrounding lipid typically has a much smaller modulus, approximately 250 mN/m [18]. While data on the area expansion modulus of proteins remain scarce, existing theoretical models for mechanosensitive channels do not support the idea that protein expansion in response to tension serves as the main gating stimulus. For the mechanosensitive Piezo channels, a dome model was proposed where tension flattens the initially curved membrane, leading to channel opening [19]. This transformation is plausible because the bending stiffnesses of the channel and bilayer are comparable [20].

Similarly, the gating of the mechanosensitive channel MscL has been linked to changes in membrane thickness. In this case, the hydrophobic mismatch between the membrane protein and surrounding lipids generates line tension, which plays a crucial role in channel gating [21].

KvAP has been described to induce membrane curvature [22], but whether channel activation alters membrane curvature has not been shown. Moreover, an adaptation of Piezo’s dome model would be questionable, as the channel area of KvAP is much smaller and much less infiltrated with lipids; hence, its bending modulus is unlikely to be as lipid-like as that of Piezo [20].

Recent cryo-electron microscopy studies of Kv channel-containing vesicles at different membrane potentials have provided structural insights into the conformational transitions associated with voltage sensor movement in the mammalian non-domain-swapped Kv channel Eag [23]. They suggested that, in the down state, the third and fourth arginine of the voltage sensor pull the interacting phosphate moieties of membrane lipids upward. Consequently, the channel resides in a membrane region approximately 5 Å thinner than its surroundings. Notably, this thinning is absent in the open channel, suggesting a relationship between membrane thickness and channel conformation. The combination of activation by membrane tension [16] and deactivation by membrane thinning [23] is counterintuitive. A possible scenario that could explain both observations is that membrane tension puts a strain on the lipid phosphates that interact with the third and fourth arginine of the voltage sensor. This hypothesis raises an intriguing question: would a mere change in bilayer thickness produce the same effect? In other words, we may hypothesize that the thickening of the membrane should also strain the phosphate–arginine interaction since pulling those interacting lipids up in the voltage sensor’s down state would require larger bilayer mechanical deformations. If so, we would expect to see an increase in KvAP channel activity upon a sudden increase in membrane thickness. To address the above, we adopted an optically excited photolipid-based approach using photolipids in solvent-depleted planar lipid bilayers (PLBs). Specifically, we utilized azobenzene-containing diacylglycerols (OptoDArG), which undergo rapid and reversible photoisomerization upon exposure to UV (375 nm) or blue (488 nm) light. The photoisomerization of membrane-embedded OptoDArG reversibly alters the mechanical properties of the lipid bilayer [24,25]. Recently, we demonstrated that this rapid photoisomerization can induce changes in membrane capacitance on the millisecond timescale, generating optocapacitive currents that lead to membrane de- or hyperpolarization in response to light [26]. Blue light (cis to trans-state) evoked hyperpolarizing optocapacitive currents by reducing the area per photolipid molecule, which was accompanied by membrane thickening and the activation of mechanosensitive channels in the plasma membrane. Our intuitive interpretation was that the rapid reduction in area per lipid upon transitioning from cis to trans photoisomers by blue light irradiation generates membrane tension (σ). Alternatively, endogenous mechanosensitive channels may have responded to isomerization-induced changes in membrane thickness, [24,27] spontaneous curvature, [25,28] or bending rigidity [29].

In this study, we first estimated the amount of σ generated by photolipid switching by imaging the movement of the anchored solvent annulus (torus) in photoswitchable PLBs, alongside high-frequency recordings of membrane capacitance. We then reconstituted KvAP, a model Kv channel with a non-domain-swapped voltage sensor, into solvent-depleted PLBs containing ~20 m% photoswitchable lipids. Photoisomerization of OptoDArG between its *cis* and *trans* conformation successfully modulated KvAP activity. Our results support the hypothesis that membrane thickening disrupts interactions between lipid phosphates and with the voltage sensor’s arginines, thereby increasing KvAP activity at a given voltage.

## 2. Materials and Methods

### 2.1. KvAP Expression and Purification

KvAP wild type was overexpressed in *E. coli* C43 (DE3) cells from a pET21a-derived expression vector, harboring a C-terminal 6xHis-tag. Cultures were grown in 2 L 2xYT media and induced with 1 mM isopropyl-β-D-thiogalactopyranoside (IPTG) for 4 h. The cell pellet was resuspended in extraction buffer (basic buffer—BB: 100 mM KCl, 25 mM Tris, pH 8.0, 1 mM MgCl₂, and one complete protease inhibitor cocktail tablet (Roche)). Cells were lysed using an Emulsiflex homogenizer (Avestin). Cell debris was removed by centrifugation at 6500× *g* and 4 °C for 10 min. The membrane fraction was pelleted by ultracentrifugation at 100,000× *g* and 4 °C for 2 h. The pellet was resuspended in BB containing 4% (*w*/*v*) n-decyl-β-D-maltopyranoside (DM, Anatrace) and 2 mM tris(2-carboxyethyl) phosphine (TCEP) for 2 h at 4 °C. Post-solubilization, the sample was centrifuged again at 100,000× *g* and 4 °C for 2 h. The supernatant was subjected to affinity chromatography using Ni-NTA agarose (Qiagen). Purified complexes were eluted with size exclusion buffer (SE: 100 mM KCl, 25 mM Tris, pH 8.0) containing 0.25% (*w*/*v*) DM and 400 mM imidazole. Elution fractions were pooled and concentrated to a final volume of 500 µL using Pierce Protein Concentrators PES, 30 kDa MWCO, at 2900–4500× *g*, and 4 °C. Size exclusion chromatography was performed on an Äkta Pure system (Cytiva, Superdex 200 Increase 10/300 GL) to enhance purity and stability and to remove aggregates. The concentration was determined by the Bradford method (BioRad, QuickStart Bradford 1x dye reagent), yielding 0.82 mg/mL for WT and 0.86 mg/mL for the variant. Aliquots of 50 µg protein were flash-frozen and stored at −80 °C for reconstitution attempts. Ni-NTA affinity chromatography fractions and SEC fractions were analyzed by loading onto a 15% SDS-PAGE gel. The purity and oligomerization state of KvAP in the presence of SDS were assessed by Coomassie staining.

KvAP was reconstituted into *E. coli* polar lipid extract vesicles (10 mg, Avanti Polar Lipids) pre-dissolved in 0.5% (*w*/*v*) DM. Protein-to-lipid mass ratios of 1:200 and 1:50 were used. Bio-Beads SM2 (BioRad) were added to remove excess detergent, followed by overnight incubation at 4 °C with gentle mixing. The suspension was then pelleted at 137,000× *g* and 4 °C for 1.5 h. The pellet was resuspended in electrophysiology buffer (EP: 10 mM HEPES, pH 7.5, 450 mM KCl, 10% (*v*/*v*) glycerol). A total of 25 µL aliquots were flash-frozen with liquid nitrogen and stored at −80 °C for subsequent experiments.

### 2.2. Horizontal Solvent-Depleted Planar Lipid Bilayers (PLBs)

Experiments followed established procedures [30]. Lipid mixture aliquots, consisting of *E. coli* Polar Lipid Extract (PLE) and OptoDArG were prepared in a glass vessel. The lipids were dissolved in chloroform, and the solvent was evaporated. The vessel was then flushed with argon gas and stored at −80 °C. Planar lipid bilayers (PLBs) were formed over apertures 70–100 µm in diameter in 25 µm thick PTFE foil (Goodfellow GmbH), created using high-voltage discharge and pretreated with 0.6% hexadecane in hexane. This process utilized a custom-built PTFE chamber assembly separating two aqueous compartments. A lipid solution (10 mg/mL in hexane) was applied as monolayers on top of the aqueous buffer in both compartments. After hexane evaporation for >1 h (except in Figure 1, where reduced evaporation time promoted a large torus), membrane folding was induced by rotating the upper chamber. Successful bilayer formation was verified via capacitance measurements, with membranes exhibiting specific capacitance values >0.75 µF/cm^2^ considered suitable for further analysis.

### 2.3. Optical Measurement and Photoisomerization

The chamber was mounted on the sample stage of an Olympus IX83 inverted microscope equipped with an iXon 897 EMCCD camera (Andor Technology Ltd, Belfast, Northern Ireland). For the photoisomerization of OptoDArG-containing PLBs, a 375 nm diode laser (iBEAM-SMART-375-S, TOPTICA Photonics) was used to induce trans-to-cis conversion, and a 488 nm diode laser (iBEAM-SMART-488-S-HP, TOPTICA Photonics) was used for *cis*-to-*trans* conversion. These lasers were focused on the back-focal plane of the objective. Controlled synchronization of the lasers, electrophysiological data acquisition, and manipulation of the motorized microscope were managed through cellSens software, version 1.11 (Olympus) utilizing a real-time controller (U-RTC, Olympus).

### 2.4. Electrophysiology Recordings

Following successful PLB folding, vesicles containing KvAP were added near the membrane, and fusion was facilitated via a salt concentration difference across the membrane. Vesicles were introduced to the hyperosmotic compartment with 150 mM KCl, while the hypoosmotic compartment contained 15 mM KCl. Both sides were buffered with 10 mM HEPES at a pH of 7.4. Voltage clamp measurements were made using an EPC 9 patch clamp amplifier (HEKA Elektronik). The current was acquired at 25 kHz and analogously filtered at 10 kHz using a combination of Bessel filters. For analysis, the currents were digitally filtered at 0.5–1 kHz. Ag/AgCl electrodes with salt bridges were used, and all measurements were conducted according to electrophysiological conventions, with the extracellular side of the membrane taken as ground. After channel appearance, both sides of the chamber were adjusted to 150 mM KCl. Capacitance recordings were obtained with a software lock-in implemented in the PatchMaster software (HEKA Elektronik). Sine wave parameters for software lock-in measurements (“Sine + DC” method with computed calibration) were 10–20 mV peak amplitude, 5 kHz frequency, 10 points per cycle, and no averaging; the DC voltage offset was ±10 mV.

### 2.5. Electrophysiology Data Analysis

Single-channel amplitude histograms were fitted with multiple Gaussians, determining the mean current and the area associated with the closed state and open states. Channel activity was calculated as the ratio of the sum of the fitted areas corresponding to channels in the open state to the sum of all fitted areas (which includes the area corresponding to the closed state). The lag timewas defined as the temporal delay to (a) the first opening of an additional channel following the complete cis-to-trans photoisomerization and (b) the first channel closing event after complete trans-to-cis photoisomerization of the PLB. The voltage dependence of channel activation, specifically the value of *V*_1/2_, was determined by fitting the normalized peak currents elicited by a range of test potentials to the Boltzmann equation (Equation (4)). Data fitting was performed on averaged datasets using the Levenberg–Marquardt algorithm. The data points of the average datasets were weighted by 1 over their respective variance. Finally, the determined *V*_1/2_ values and other fit parameters are reported as mean ± standard error of the fit. All data fitting and graphing were performed in Origin 2024 (OriginLab).

## 3. Results

### 3.1. Fast Photoisomerization of OptoDArG Generates Only a Small Transient Increment in Membrane Tension

To investigate whether photolipid isomerization significantly augments membrane tension (*σ*), we folded planar lipid bilayers (PLBs) with a prominent anchoring solvent annulus (Figure 1a). While typically minimized to reduce residual solvent [31] (yellow in Figure 1e), the larger annuli in our experiments enabled visualization of the torus’s inward deformation under intense blue light (Figure 1b). Initially, UV light induced the cis state of the bilayer, i.e., a bilayer state in which most photolipids are in their cis conformation, characterized by finite membrane tension, *σ*_0_, and capacitance, $C_{m}$ (**1** in Figure 1c,d). Upon equilibrium, blue light exposure triggered cis-to-trans photoisomerization, leading to a rapid reduction in bilayer area ($A_{m}$, Appendix A). This increased *σ*, resulting in a reshaped torus (Figure 1e). The origin of *σ* is trans-OptoDArG’s smaller molecular area compared to its cis counterpart (Figure 1f).

A rough estimation of *σ* can be made from the observed radius change. Immediately after blue light exposure, the radius of the bilayer in the middle of the toroidal reservoir decreases from *r*_0_= 29.5 µm to *r_in_* = 27.5 µm (Figure 1b) before quickly relaxing to its initial value (within one frame). Importantly, key parameters depend on ln(*r*_0_/*r_in_*), making the exact values less critical. Denoting the equilibrium lateral tension of the membrane at *r*_0_ as *σ*_0_ and the tension at *r_in_* as *σ*_0_(1 + α), we assume that the lateral tension evolves linearly with time, *t*, during the relaxation (0 < *t* < *τ*). Thus, *σ*(*t*) = A*t* + B, with boundary conditions *σ*(0) = *σ*_0_(1 + α) and *σ*(*τ*) = *σ*_0_, allowing us to determine A and B as A = –α*σ*_0_/*τ* and B = *σ*_0_(1 + α).

The radial flow of viscous lipid follows the following equation [32]:(1)$$4\pi \eta h\dot{r}=2\pi r\sigma$$ where *η* is the surface viscosity (membrane viscosity times membrane thickness).

When *σ* = *σ*_0_, the membrane rim is in equilibrium (i.e., no motion occurs). Substituting *σ*(*t*) − *σ*_0_ = α*σ*_0_ (1 − *t*/*τ*) into Equation (1) and solving for *r*(*t*) with boundary conditions *r*(0) = *r_in_* and *r*(*τ*) = *r*_0_ yields the following solution:(2)$$r\left(t\right)=r_{in}\exp\left[\ln\left(\frac{r_{0}}{r_{in}}\right)\frac{\left(2\tau -t\right)t}{\tau^{2}}\right]$$

Notably, *r*(*t*) changes exponentially, as do $A_{m}$ and $C_{m}$ (Figure 1c), supporting the assumption of a linear dependence of σ on *t*. To estimate *τ*, we need to estimate the rate at which σ decays following blue light exposure. To that effect, we obtained consistent results from imaging and $C_{m}$ measurements.

From the video recordings, we quantified the torus movement by averaging 11-line profiles along the equator of the PLB (yellow lines, Figure 1a), resulting in an average profile $Z_{i}$ for each frame i. For seven instances of cis-to-trans photoisomerization (see Appendix A), we selected $Z_{i}$ just before light exposure and verified that it was indistinguishable from the previous profile $Z_{i-1}$ by calculating the difference $Z_{i}-Z_{i-1}$ (black line in Figure 1b). Following blue light exposure, $Z_{i}-Z_{i+1}$ (blue line in Figure 1b) displayed pronounced positive ($r_{\mathrm{in}}$) and negative ($r_{0}$) deflections, indicating inward torus movement from $r_{0}$ to $r_{\mathrm{in}}$. Finally, $Z_{i}-Z_{i+2}$ (red line in 1b) no longer exhibited these peaks, showing that the torus relaxes within less than twice the frame interval of 40 ms, i.e., τ<80 ms.The rate of decrease in $C_{m}$ at the onset of blue light exposure is rapid and largely determined by irradiance and azobenzene photoisomerization kinetics [26]. $C_{m}$ reaches a minimum as OptoDArG is quantitatively photoisomerized (**2** in Figure 1c), resulting in bilayer thickening and a reduction in $A_{m}$ as the torus is pulled inwards. Then, $C_{m}$ relaxes with an estimated time constant τ≈23 ms (monoexponential fit to the relaxation in $C_{m}$; red line in Figure 1c). This increase in $C_{m}$ is consistent with the observed increase in $A_{m}$ as the torus retracts. Hence, we conclude that the relaxation in torus position and $C_{m}$ reflect the same underlying phenomenon.

With the above-determined values *r*_0_ = 29.5 µm, *r_in_* = 27.5 µm, and *τ* = 23 ms and assuming the tension of a planar bilayer to be *σ*_0_ ≈ 4 mN/m [33], we can find α by solving Equation (2) with the boundary conditions imposed above on *r*(*t*).(3)$$\alpha =4ln\left(\frac{r_{0}}{r_{in}}\right)\frac{\eta h}{\sigma_{0}\tau }$$

For solvent-depleted PLBs, we previously measured ηh = 4 nNs/m [30]. Substituting this value into Equation (3) yields an increment in *σ* of 0.001%. Yet estimates for ηh vary considerably—typically between 0.15 and 15 nNs/m [34]. For lipid domains, values as high as 500 nNs/m have been reported [35]. Using this upper value, we estimate α≈0.15%, suggesting that the tension could increase from 4 to 4.006 mN/m. However, there is no previous evidence indicating that Kv channels can respond measurably to such tiny changes in tension.

### 3.2. Functional Reconstitution of KvAP into Solvent-Depleted PLBs Containing OptoDArG

KvAP was expressed in Escherichia coli and purified following a protocol previously established for the bacterial K^+^ channel KcsA [36]. Extraction from the cell pellet using n-dodecyl-β-D-maltoside (DDM) was validated via size-exclusion chromatography (SEC) and SDS-PAGE analysis. The SEC chromatogram exhibited a peak at an elution volume of 11 mL, consistent with the expected size of tetrameric KvAP (e.g., ref [37]). SDS-PAGE further confirmed the tetrameric assembly of the channel in detergent micelles (Figure 2a).

Subsequently, DDM was replaced with E. coli polar lipid extract (PLE), forming vesicles that incorporated KvAP into their membranes. These vesicles were fused with solvent-depleted planar lipid bilayers (PLBs) composed of PLE (specific capacitance ≥ 0.75 µF/cm^2^) containing 0 or 20 m% OptoDArG. Fusion was driven by a 150 to 15 mM KCl gradient [36]. This method facilitated the study of single-channel and ensemble currents by varying the protein-to-lipid ratio.

The single-channel conductance of KvAP (Figure 2d) recorded in photoswitchable PLBs containing 20 m% trans-OptoDArG in 150 mM KCl was determined to be 190 ± 6 pS, based on the single-channel current–voltage (I/V) relationship (Appendix A). This value closely aligns with previously reported conductance measurements [38]. To assess the functionality of KvAP by ensemble current recordings, we fused proteoliposomes containing a high protein-to-lipid ratio to PLBs, both with and without OptoDArG. Under both conditions, KvAP exhibited robust voltage-dependent gating (Figure 2b,c,e). Channel activity was inhibited by 0.4 µM of charybdotoxin (CTX), a known pore blocker of KvAP (Figure 2c). This inhibition follows a key-lock mechanism described in previous studies [38,39,40]. Voltage-dependent activation and CTX-mediated inhibition confirm the successful reconstitution of functional KvAP channels into photoswitchable PLBs.

**Figure 2 biomolecules-15-00744-f002:**
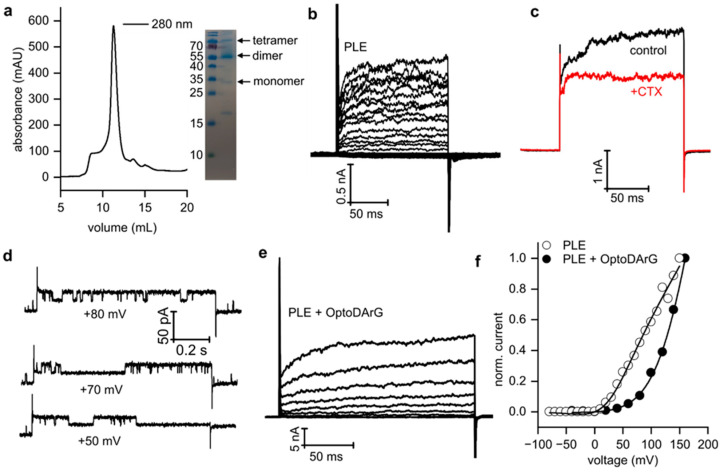
Functional reconstitution of KvAP into solvent-depleted PLBs. (**a**) (Left) Size-exclusion chromatogram showing a peak at 11 mL, corresponding to KvAP. (Right) SDS-PAGE analysis confirming KvAP tetramers in detergent micelles. Original images can be found in Appendix A.(**b**) Voltage clamp current recordings on a solvent-depleted PLB containing reconstituted KvAP channels. Voltage pulses were applied from −80 mV to +150 mV in 10 mV increments, with a holding potential (h.p.) of −100 mV and 5 s inter-sweep delays. (**c**) Inhibition of KvAP by 0.4 µM CTX. Currents were recorded during voltage pulses from −80 mV to +100 mV, with an h.p. of −100 mV. (**d**) Single-channel recordings of KvAP in photoswitchable PLBs. (**e**) Current recordings from a photoswitchable PLB. Voltage pulses were applied from −40 mV to +160 mV in 20 mV increments, with an h.p. of −100 mV and 5 s inter-sweep delays. (**f**) Normalized I/V curves constructed from the records shown in (**b**) (open circles) and (**e**) (closed circles), fitted using Equation (4). Panels (**b**–**f**): PLBs were folded from PLE with and without 20 m% OptoDArG in 150 mM KCl, 10 mM HEPES, pH 7.4.

### 3.3. Voltage-Dependent Gating of KvAP Is Influenced by Solvent Content and the Presence of Non-Phospholipids

I/V curves derived from ensemble recordings in the absence (Figure 2b) and presence of trans-OptoDArG (Figure 2e) were fitted by a Boltzmann equation as follows: [41](4)$$I\left(V\right)=\frac{\left(V-V_{rev}\right)\times G_{max}}{1+\exp\left(\frac{V-V_{1/2\;}}{V_{s}}\right)}$$ where *V* represents voltage, and the parameters *V_rev_*, *G_max_*, *V*_1/2,_ and *V_s_* correspond to the reversal potential, maximum conductance, half-maximal activation voltage, and slope factor, respectively.

In PLBs composed of PLE, a pronounced depolarizing shift in *V*_1/2_ to 28 ± 5 mV (Figure 2b,f) was observed compared to earlier studies using black lipid membranes (BLMs) painted from 1:3 POPE:POPG in n-decane, where *V*_1/2_ was −51 mV [38]. This shift likely arises from decane retained between the leaflets in BLMs, which is also reflected in their lower specific capacitance compared to PLBs. Our findings align with prior studies [42,43] showing a large difference in *V*_1/2_ for Kv1.2 channels, transitioning from −70 mV in BLMs to −3.5 mV in oocytes. Similarly, KvAP studies in other solvent-free systems, such as giant unilamellar vesicles (GUVs) [37] and bead-supported unilamellar membranes (bSUMs) [44], also revealed a positive shift in voltage activation.

Furthermore, the incorporation of 20 m% OptoDArG into the PLBs produced an additional rightward shift in *V*_1/2_ (Figure 2f). This indicates that 20% of the non-phospholipid OptoDArG (a diacylglycerol) modulates KvAP activation, emphasizing the necessity of phosphate groups for Kv gating [13]. Additionally, we observed that channel activation kinetics in PLBs were slower compared to BLMs (Figure 2e). These characteristics are consistent with other studies of Kv channels, including the paddle chimera and KvAP, in solvent-free bilayers such as oocytes, GUVs, and bSUMs [42,45,46]. In our experiments, the observation time was too short to resolve the inactivation process, which occurs on a longer timescale [38].

In summary, the comparison of KvAP function in solvent-depleted PLBs and other solvent-free bilayers with decane-containing BLMs suggests that residual solvent intercalated within the membrane affects KvAP voltage-dependent behavior. Also, the presence of OptoDArG, which lacks a phosphate group, exerts a pronounced effect on voltage-dependent channel function.

### 3.4. Rapid Photoisomerization of Membrane-Embedded OptoDArG Gates KvAP

Following the functional characterization of KvAP in PLBs, we investigated how changes in thickness influence KvAP gating. The single-channel activity of KvAP was modulated by the photoisomerization of OptoDArG (Figure 3a). Experiments began with a 40 ms UV light pulse, converting all OptoDArG molecules into their cis configuration. This transition reduced membrane thickness, leading to an increase in membrane capacitance ($C_{m}$) [26]. The associated optocapacitive current was characterized by a rapid, transient depolarizing peak, followed by a decrease in the activity of KvAP (Figure 3a–c). We observed that the number of active channels was different in the cis and trans states of the PLB. Hence, the fraction of open channels, in the following called channel activity, is increasing.

Subsequent illumination with a 40 ms blue light pulse induced the reversible isomerization of OptoDArG to its trans conformation. This transition decreased C_m_ and the area per photolipid (Figure 1), resulting in a persistent increase in membrane thickness. The thicker membrane favored KvAP channel openings, as indicated by a higher activity (Figure 3a–c).

Exposure to another UV light pulse reduced KvAP channel activity again, confirming the reversible modulation of KvAP gating by photolipid photoisomerization and the associated changes in bilayer properties. Importantly, no significant difference in KvAP single-channel conductance was observed between cis and trans-PLB states; in both cases, the conductance value agreed with measurements in the absence of OptoDArG.

Control experiments without KvAP channels demonstrated that UV and blue light elicited optocapacitive currents without affecting bilayer permeability (Figure 3a), thereby ruling out photolipid-induced artifacts. Additional control measurements demonstrated that KvAP is not inherently photosensitive (Appendix A). Notably, we observed a temporal lag in the KvAP single-channel response to photoisomerization (Figure 3d). Upon blue light exposure, the bilayer as an ensemble switched to the photostationary, mostly trans configuration within 1–2 ms (Figure 1c); however, the channel responses were detected only after a delay of approximately 10 ms. In contrast, during the reverse transition from trans to cis upon UV light exposure, channel closure required around 40 ms.

**Figure 3 biomolecules-15-00744-f003:**
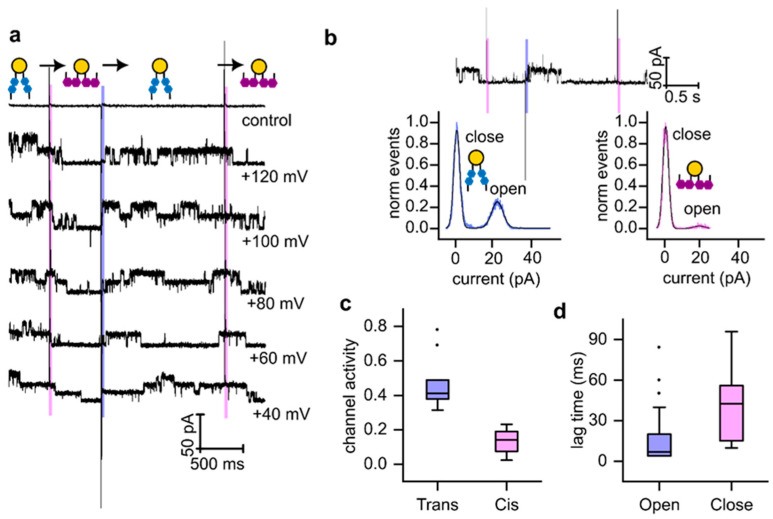
Modulation of KvAP single-channel activity by rapid photoisomerization of OptoDArG. (**a**) Representative single-channel recordings from a planar lipid bilayer (PLB) containing 20 m% OptoDArG and reconstituted KvAP channels. Recordings were conducted in 150 mM KCl, 10 mM HEPES, and pH 7.4. Photoisomerization of OptoDArG was induced by 40 ms pulses of UV (magenta bars) and blue (blue bars) laser light. (**b**) Amplitude histograms of KvAP single-channel activity, constructed from the recording shown above. These histograms were used to calculate the activity by determining the fraction of time at least one of the channels was in the open state. (**c**) Box plots illustrating channel activity at an applied voltage of +120 mV for trans-OptoDArG (blue) and cis-OptoDArG (magenta). Data represent seven independent experiments for each photoisomer. (**d**) Lag times between OptoDArG photoisomerization and changes in channel activity: (blue) lag between cis-to-trans photoisomerization and first channel opening; (magenta) lag between trans-to-cis photoisomerization and first channel closing. Lag times were calculated from 33 channel-opening events and 34 channel-closing events.

Our single-channel experiments demonstrated that light can be used to increase KvAP channel activity. While the observed changes in activity were modest overall, with an average increase of approximately 30%, certain experiments revealed much more pronounced effects. A striking example is shown in (Figure 4), where channel activity increased from ~5% to 100%. Initially, only one channel was active when OptoDArG was in its cis state. Upon switching to the trans state, five additional channels opened, leading to a persistent increase in conductance that remained elevated throughout the observation period. The additional channels cannot be recruited from the torus because (a) vesicle fusion delivers them exclusively to the bilayer, and (b) hydrophilic loops likely prevent their migration to the torus. When OptoDArG was switched back to its cis state, all channels but one closed after a short lag period. This variability in channel behavior complicates obtaining reliable statistics for the light-induced effects.

One consistent observation emerged from every single experiment: channel activity remained elevated for a much longer duration than it took for membrane capacitance to relax after the photolipid was switched to its trans state. From this, we conclude that tension-related effects cannot be responsible for the increase in activity, as tension would be expected to relax with the same time constant as membrane capacitance, i.e., within 23 ms. However, the interval between blue and UV illumination during which elevated activity was observed exceeded 1 s. This observation aligns well with the calculated increment in tension, which was determined to be too small to significantly influence activity.

**Figure 4 biomolecules-15-00744-f004:**
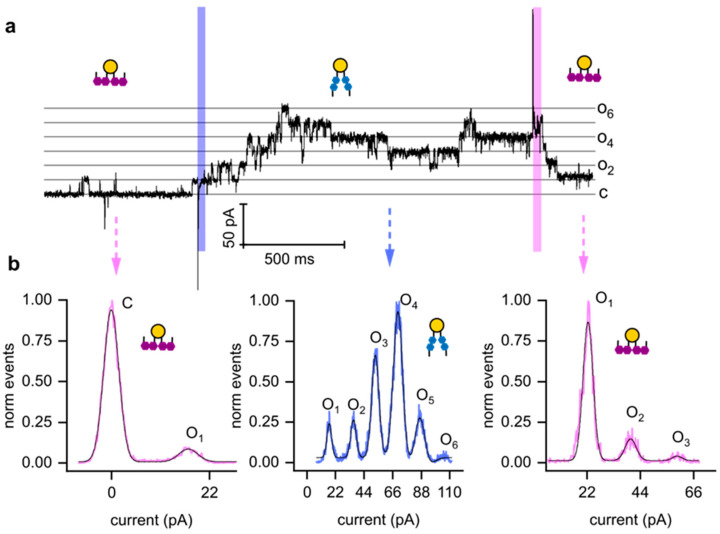
Effective modulation of KvAP channel activity by photoisomerization. (**a**) Opening and closing of KvAP channels induced by illumination with blue and UV light at +100 mV. (**b**) Amplitude histograms constructed from the recording shown in panel (**a**). The experimental conditions were similar to those described in Figure 3, except for the reconstitution of a higher number of KvAP channels.

We further conducted measurements of light-modulated KvAP conductance at the ensemble level to reduce variability and improve statistical robustness (Figure 5a). Steady-state currents recorded after the relaxation of tension under cis and trans isomer conditions (i.e., >23 ms after illumination, as shown in Figure 1) demonstrated that in the trans-state, where the membrane is thicker, the current was increased. This indicates that blue light induces the opening of some channels. The corresponding activity increase resulted from a leftward shift in *V*_1/2_ from 158.6 ± 15.8 mV (mean ± standard error, *n* = 3) in the cis state to 120.9 ± 8.3 mV (mean ± standard error, n = 3) in the trans state of the PLB (Figure 5b), qualitatively consistent with the activity changes observed in our single-channel measurements (Figure 3).

This current modulation was reversible, as the initial current was restored by UV illumination (Figure 5c). Specifically, in the cis state, where the membrane is thinner, a decrease in current was observed. These results confirm that photoisomerization of OptoDArG enables reversible control of KvAP activity through alterations in bilayer properties. Notably, a modest change in capacitance of approximately 6%—indicating an equally modest change in membrane thickness—resulted in a disproportionally larger increase in conductance (~30%).

**Figure 5 biomolecules-15-00744-f005:**
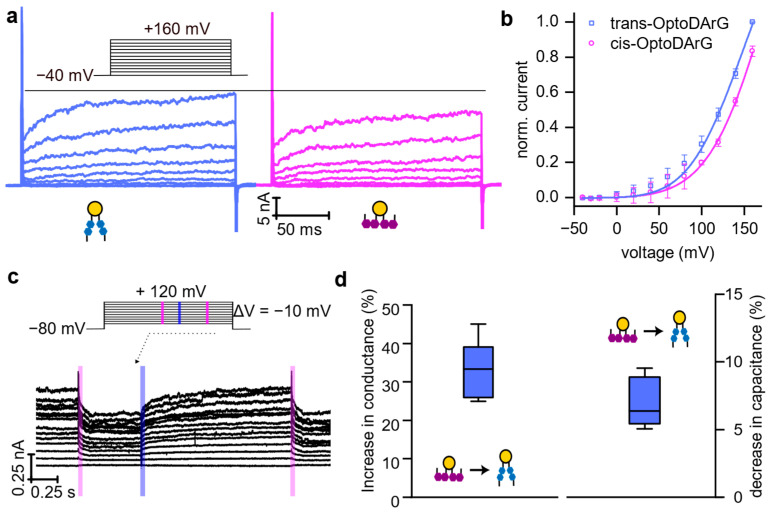
Modulation of KvAP activity by photoisomerization of OptoDArG at the ensemble level. (**a**) Voltage-dependent current recordings on a KvAP-containing photoswitchable PLB under the same experimental conditions as in Figure 3. The blue and magenta traces indicate recordings after membrane exposure to blue and UV laser light, i.e., in the presence of OptoDArG in its trans and cis state, respectively. Voltage was stepped in 20 mV increments, with pulses ranging from −40 mV to +160 mV. The inter-sweep delay was 5 s. (**b**) Average I/V curves generated from the recordings in panel (**a**) (same color code) and two other equally conducted experiments. A Boltzmann function (Equation (4)) was fitted to the data (solid lines). (**c**) The photoinduced increase or decrease in KvAP ensemble current (light exposure indicated by blue and magenta stripes) is reversibly regulated by light. The elevated current observed before the first UV pulse reflects channel activity in the trans-state of the PLB, resulting from prior exposure to blue light. Voltage was stepped in 10 mV increments, with pulses ranging from −80 mV to +120 mV. (**d**) Left panel: A box plot showing the relative increase in conductance upon photoisomerization from cis to trans-OptoDArG, calculated from the experiments in panels (**a**,**c**), and their three repetitions. Right panel: A box plot showing the relative decrease in steady-state capacitance upon cis- to trans-OptoDArG photoisomerization and in the absence of KvAP (three independent experimental runs, see Figure 1).

## 4. Discussion

We demonstrate that light-induced changes in membrane thickness enhance the activity of the model Kv channel KvAP. This observation suggests that in the up-state of the voltage sensor, interactions between the voltage-sensing arginines and the phosphate moieties of the lipid membrane are weakened. Because these interactions depend on elastic deformations of the lipid bilayer, sustaining them in thicker membranes would require more pronounced deformations than in thinner ones (Figure 6). Displacement of charged lipids may influence the adjacent electric double layer. However, the associated energetic cost is minimal, with an estimated change of less than 1 kT under physiological conditions (see Appendix A).

Supporting evidence for VSD–lipid interactions is provided by cryo-electron microscopy studies on the Eag channel, which—like KvAP—is a non-domain-swapped Kv channel. In the down-state, the voltage sensor of Eag was found to interact with negatively charged phosphate headgroups, specifically POPG lipids in reconstituted liposomes, pulling them upward and leading to a localized membrane thinning [23]. Importantly, these structures were obtained under applied membrane voltage, a condition that favors downward movement of the S4 helix.

In contrast, the recent structures of Eag reported by Zhang et al. [47] were determined at 0 mV, i.e., in the absence of membrane potential. It is therefore not surprising that no significant movement of the S4 segment was observed. Although the authors describe the pore as closed, based on its structural conformation, the voltage sensor is unlikely to reflect a fully closed (resting) state.

Notably, we found good structural alignment between the non-domain swapped Kv channels Eag and KvAP (Appendix A), indicating that the VSD-lipid interactions should also be relevant for KvAP. In thicker membranes, the arginines cannot exert the same control over lipid positioning, as pulling the lipids over a longer distance—and thus inducing greater membrane deformation—is energetically more costly. Importantly, we used *E. coli* lipid in our experiments, which contains negatively charged phospholipids, enabling such electrostatic interactions at the inner leaflet. The resistance to membrane bending is further increased by the fact that membranes containing trans-OptoDArG are stiffer [25]; i.e., they offer more resistance to bending than membranes containing cis-OptoDArG.

Additional support for the involvement of the VSD in photolipid-mediated gating of KvAP comes from a comparison with the bacterial potassium channel KcsA. While membrane thinning increases KcsA activity by reducing lateral pressure and facilitating gate opening [48], it decreases KvAP activity (Figure 3a). This opposite behavior likely arises from the presence of the VSD in KvAP. According to our model, the interaction between the VSD and the surrounding lipids differs between the up and down states of the VSD. These state-dependent interactions explain why the channel’s activity depends on membrane thickness. Moreover, the VSD also shields the central pore domain (S5–S6) from direct lipid-mediated forces.

Since the KvAP channel has a non-cylindrical shape, it is thought to sit at the bottom of a funnel-like deformation formed by the surrounding lipids [22]. Consequently, a photo-induced change in bending rigidity would be expected to result in a shallower funnel, a transformation that may exert strain on the protein and thus contribute to gating. However, such effects can be excluded under our experimental conditions; PLBs typically exhibit membrane tensions of ~ 4 mN/m [33] and according to (Figure 2) in Reference 22, the diffusion coefficient of KvAP reaches a maximum at this tension. This value corresponds to what is expected in the absence of a funnel-shaped deformation. Because this maximum is observed independently of whether the photo-lipid is in the cis or trans state, the funnel shape is likely flattened in both cases.

**Figure 6 biomolecules-15-00744-f006:**
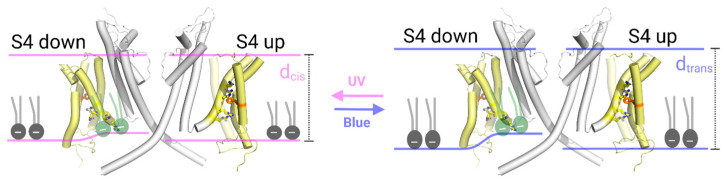
Photolipid photoswitching increased the activity of voltage-sensitive potassium channels. **Left panel**: OptoDArG is in its cis state. A subunit in the down conformation (left) is compared to a subunit in the up conformation (right), with the other subunits omitted for clarity. The voltage sensor domains (yellow) and the pore domains (gray) are visualized using PyMOL, based on cryo-EM structures of the Eag channel embedded in liposomes (PDB entries: 8EP1 and 8EOW), which capture the channel in its down and up states, respectively. The approximate position of the membrane is indicated by pink lines. The phenylalanine marking the gating charge transfer center is shown in orange [49]. In the closed state, the membrane is approximately 5 Å thinner at the site where phosphate interacts with the third and fourth arginine residues [23]. Consequently, elastic deformations must occur at these indentation sites. The lipids being pulled up by the arginines are illustrated manually. As the charged arginines of the sensor rotate toward the protein in the up conformation, they can no longer interact with the oppositely charged lipid moieties. Consequently, the lipid deformations found in the down state are absent in the up state (right schematic of the left panel). **Right panel**: Conversion of OptoDArG to its trans state leads to membrane thickening. To maintain the interaction between the arginine residues and the phosphate groups, the cytoplasmic leaflet would need to be pulled up by ~8 Å. However, the resulting lipid deformations are too large to be compensated by the interaction energy. Consequently, the interaction between the arginines and the phosphate becomes energetically less favorable, preventing the large-scale lipid deformation depicted in the left scheme. As a result, less voltage is required to drive the upward movement of the voltage sensor, shifting the current–voltage relationship to the left. The color scheme and PDB entries remain consistent with left panel.

Given the strong conservation of voltage sensor structure across Kv channels, light-induced membrane thickness changes may provide a general mechanism for reversibly regulating Kv channel activity in the presence of photoswitchable lipids. Importantly, this mode of regulation is particularly versatile, as it exploits a fundamental physical property—membrane thickness—rather than requiring direct modifications to the channel itself.

## 5. Conclusions

Our findings demonstrate that an increase in membrane thickness destabilizes the S4 segment of the VSD in its down position. Consequently, the upward movement of S4 is facilitated, leading to increased activity of KvAP. These changes manifest as a leftward shift in the current voltage curve and may activate previously silent channels. By inducing these changes optically, our study paves the way for using photolipids as a novel approach to regulate neuronal excitability.

## Figures and Tables

**Figure 1 biomolecules-15-00744-f001:**
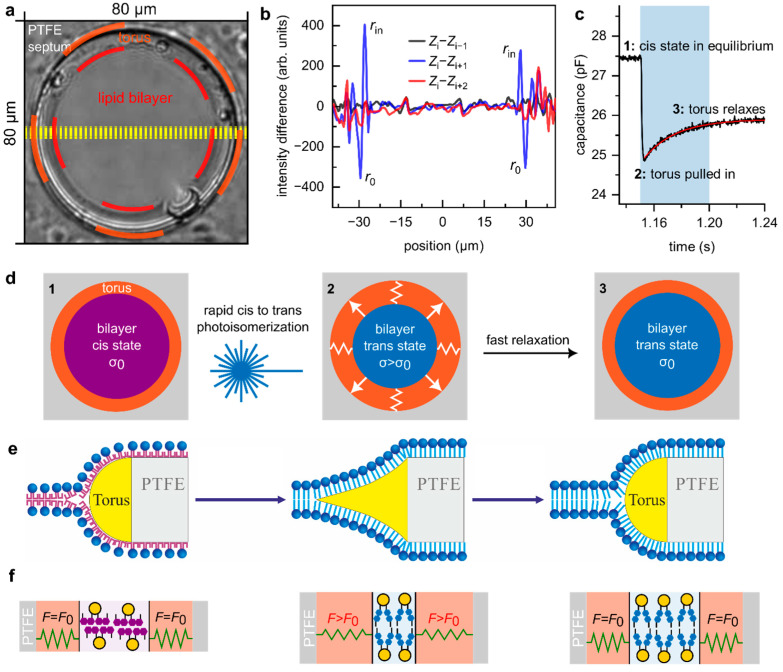
The increment in bilayer tension due to lipid photoisomerization is negligible. (**a**) A transmitted light image of a photoswitchable PLB with an enforced large torus. Lipid bilayer, torus, and PTFE septum are indicated. (**b**) $Z_{i}$ corresponds to the average line profile (obtained from the yellow dashed lines in panel (**a**) of frame i immediately before blue light exposure, $Z_{i-1}$ is the line profile of the frame before, $Z_{i+1}$ and $Z_{i+2}$ are the average line profiles corresponding to one and two frames after frame i, respectively. The calculated differences demonstrate the inwards movement of the torus following blue light exposure ($Z_{i}-Z_{i+1}$, blue line) and the return to the initial shape one frame later ($Z_{i}-Z_{i+2}$, red line). The lines are averaged from seven switching events on the same PLB. (**c**) Time course of $C_{m}$ upon blue light exposure (indicated by the blue background) of a cis-OptoDArG containing PLB. $C_{m}$ decreases rapidly as photolipids switch to the trans state and the torus is pulled inwards; subsequently, $C_{m}$ relaxes, which is partially due to the increase in $A_{m}$ as the torus retracts again. The red line is a monoexponential fit to the relaxation. (**d**) Schematic view of the response of a photoswitchable PLB to blue light irradiance. σ may be increased transiently as the torus is stretched by the bilayer. As described below, σ relaxes within 80ms. (**e**) Schematic representation of torus shape corresponding to the bilayer states in panel (**d**). (**f**) Schematics of how photoisomerization from cis- to trans-OptoDArG generates tension within the membrane. The gray bars labeled PTFE symbolize the septum aperture within which the PLB is mounted, and green springs refer to the bilayer lipids other than OptoDArG.

## Data Availability

The research data are available on https://doi.org/10.5281/zenodo.15465224.

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
