# Peer review of "Modulation of Kv Channel Gating by Light-Controlled Membrane Thickness"

_biomolecules, 2025, doi:10.3390/biom15050744_

Round 1

Reviewer 1 Report

Comments and Suggestions for Authors

The manuscript describes an increase in activity of the voltage-gated channel KvAP expressed in a lipid bilayer containing the photoswitchable lipid OptoDArG when the thickness of the bilayer is increased by light-induced isomerization. The results are interesting and provide valuable information. I have only some minor remarks.

-A demonstration that the changes observed require the presence of OptoDArG, that is, that the channel itself is not photosensitive, would be crucial to support the conclusions.

-In Figure 5, please explain the high activity before the UV illumination and isomerization. Was the bilayer exposed to blue light before?

-Line 338. The statement “inactivation was incomplete” seems an understatement; it is absent in the time displayed. Rute et al. described inactivation using much longer stimuli.

-Lines 471-472. What is the evidence that the funnel shape is flattened “regardless of the state of the photolipid?

-I could not find the supplementary video mentioned in line 211.

Methods:

-It looks like the channel carries a His-tag (since it is purified by nickel chelation). This should be explicitly stated.

-Consistent use of relative centrifugal force instead of rpm would be advisable.

-Please clarify: did you use Ag/AgCl electrodes or salt bridges?

Reviewer 2 Report

Comments and Suggestions for Authors

In this study, Rohit Yadav et al used reconstituted photoswitch lipids with planar lipid bilayer to achieve light-controlled regulation of Kv channel gating. They find the activity of the KvAP channel was increased ~30% upon membrane thickening induced by blue light. While UV light induces thinning membrane and decrease the KvAP channel activity. Although this study presents an interesting concept and demonstrates modulation of Kv channel activity by altering membrane thickness, I think further investigation is needed to fully support their novel finding.

  1. The author only used the cryo-em structures of Eag Kv channel to show that the arginine residues of the voltage sensor draw lipid phosphates upward, leading to a local membrane thinning of ~5 Å. While the overall structure of Eag Kv is different from that of KvAP. It is better to make some structure comparison or sequence alignments to prove that KvAP also has a conserved arginine residues to induce the membrane thinning like KvAP. Will it possible to use alphafold 3 to predict the interactions between lipid phosphates and the voltage-sensing arginines of KvAP?

  1. Interestingly, I think author could discuss the difference between the membrane tension on KvAP and kcsA (https://doi.org/10.1080/19336950.2019.1676367). In thin membrane, the membrane leaflets distort to accommodate hydrophobic mismatch. The reduced lateral pressure enables KcsA gate to open more readily. In contrast, the thinning membrane tends to close the KvAP gate and reduce the channel activity of KvAP. It is better to compare with their finding with the exist KcsA study.

  1. The author used the SDS-PAGE gel to show that they purified the correct KvAP protein in Figure 2a, in their SDS-PAGE gel, the ratio of KvAP tetramer is lower than that of KvAP dimmer. But the size exclusion shows a single peak. The author can explain why KvAP tetramer dissociate in the SDS-PAGE gel. Will the monomer or dimer form of KvaP affect the channel activity of KvAP?

  1. In line 100, may change the sentence “by blue light irradiation generates membrane tension, 𝜎.” to “by blue light irradiation generates membrane tension (𝜎).”

Reviewer 3 Report

Comments and Suggestions for Authors

The article investigates how voltage-gated potassium (Kv) channel gating is affected by light-induced changes in membrane thickness. Authors have found that KvAP channel interactions with anionic lipids play a significant role in its gating mechanisms. Authors state that when the channel is in its down state, certain arginine residues contribute to local thinning of the membrane; therefore, thickening the membrane (with optoswitchable phospholipids) can increase KvAP activity.

The described mechanism offers a non-genetic approach to control ion channel activity using light; however, it cannot guarantee selectivity of action on specific ion channels in mixed systems like living cells.

The described mechanism provides a hint about the possible positive modulation of channel activity within lipid rafts, which are characterized by a large membrane thickness.

Questions:

(1) Line 156. Authors state that lipid layers exhibiting capacitance values >0.75 μF/cm² are considered suitable for further analysis. Why this value and how does it relate to membrane thickness?

(2) How does the thickness of the model bilipid layer compare with the actual membrane thickness of cells? To what extent (in angstroms or in %) does the insertion of 20 m% OptoDArG change the membrane thickness in cis and trans conformations?

Reviewer 4 Report

Comments and Suggestions for Authors

see attached file
